# Multispectral Remote Sensing as a Tool to Support Organic Crop Certification: Assessment of the Discrimination Level between Organic and Conventional Maize

Antoine Denis [1,*], Baudouin Desclee [2], Silke Migdall [3], Herbert Hansen [2], Heike Bach [3], Pierre Ott [4], Amani Louis Kouadio [1,5] and Bernard Tychon [1]

1  Water, Environment and Development Unit, Environmental Sciences and Management Department, Arlon Campus Environment, UR SPHERES, University of Liège, 185 Avenue de Longwy, 6700 Arlon, Belgium; Louis.Kouadio@usq.edu.au (A.L.K.); Bernard.Tychon@uliege.be (B.T.)
2  KEYOBS SA, CAP Business Center, 31 Rue d'Abhooz, 4040 Herstal, Belgium; baudouin.desclee@ec.europa.eu (B.D.); HHansen@odinix.be (H.H.)
3  VISTA GmbH, Gabelsbergerstraße 51, D-80333 München, Germany; Migdall@vista-geo.de (S.M.); Bach@vista-geo.de (H.B.)
4  ECOCERT SA, BP 47, Lieu dit Lamothe, 32600 L'Isle Jourdain, France; Pierre_R_Ott@orange.fr
5  Centre for Applied Climate Sciences, University of Southern Queensland, West Street, Toowoomba, QLD 4350, Australia
*  Correspondence: Antoine.Denis@uliege.be

**Abstract:** The annual certification of organic agriculture products includes an in situ inspection of the fields declared organic. This inspection is more difficult, time-consuming, and costly for large farms or in production regions located in remote areas. The global objective of this research is to assess how spatial remote sensing may support the organic crop certification process by developing a method that would enable certification bodies to target for priority in situ control crop fields declared as organic but that would show on satellite imagery an appearance closer to conventional fields. For this purpose, the ability of multispectral satellite images to discriminate between organic and conventional maize fields was assessed through the use of a set of four satellite images of different spatial and spectral resolutions acquired at different crop growth stages over a large number of maize fields (32) that are part of an operational farm in Germany. In support of this main objective, a set of in situ measurements (leaf hyperspectral reflectance, chlorophyll, and nitrogen content and dry matter percentage, crop canopy cover, height, wet biomass and dry matter percentage, soil chemical composition) was conducted to characterize the nature of the biochemical and biophysical differences between organic and conventional maize fields. The results of this research showed that highly significant biochemical and biophysical differences between a large number of organic and conventional maize fields may exist at identified crop growth stages and that these differences may be sufficiently pronounced to enable the complete discrimination between crop management modes using satellite images issued from quite common multispectral satellite sensors through the use of spectral or spatial heterogeneity indices. These results are very encouraging and suggest, for the first time, that satellite images could effectively support the organic maize certification process.

**Keywords:** spatial remote sensing; multispectral satellite image; organic crop certification; organic agriculture; conventional agriculture; maize; biochemical and biophysical maize properties; discrimination

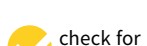



## 1. Introduction

Organic agriculture is continuously developing globally, with an expansion of 400% for land area and 500% for the market between 1999 and 2014 [1]. Organic agriculture products are in increasing demand, principally because, when compared to conventional products, organic food is seen to be healthier and organic farming is more respectful of the environment by avoiding the use of synthetic chemical fertilizers and pesticides.

In most countries with a developed and substantial organic market, organic farming products must comply with established international standards and rules in order to be labelled organic for sale, and organic producers have to be certified annually by organic certification bodies. Part of this annual certification process consists of an in situ inspection of the fields declared organic.

The global objective of this research is to assess how spatial remote sensing may support the organic crop certification process by developing a method that would enable certification bodies to target for priority in situ control crop fields declared as organic but that show an appearance on satellite imagery closer to conventional fields. For this purpose, the ability of multispectral satellite images to discriminate between organic and conventional maize fields was assessed through the use of a set of satellite images of different spatial and spectral resolutions acquired at different crop growth stages over a large number of maize fields (32) being part of an operational farm in Germany. In support to this main objective, a set of in situ measurements was conducted to characterize the nature of the biochemical and biophysical differences between organic and conventional maize fields. In particular, hyperspectral (350–2500 nm) reflectance measurements were used to identify the wavelengths and their combinations enabling the best discrimination between crop management modes.

The methodology developed in this research is expected to primarily support organic crop certification bodies by providing them with a supplementary and independent means of control of the organic nature of the fields to be certified. This would be particularly useful for large farms or in production regions located in remote areas where the control and certification process might be more difficult and costly. Additionally, this method may also interest public authorities as an additional means to supervise the proper functioning and the efficiency of the organic crop certification bodies.

This research is based on the hypothesis that the differences in treatments between organically and conventionally managed crop fields, primarily the differences in fertilization and crop protection, would result in biochemical and biophysical differences between the fields. These possible differences for organic crops compared to conventional ones were assumed to be: (i) a lower crop biomass development due to the lower organic fertilization and the lower crop protection, (ii) a lower nitrogen and chlorophyll content due to the lower organic fertilization, and (iii) a higher field spatial heterogeneity due to a higher expression of the spatial heterogeneity of the natural fertility potential of a field that is less compensated by the organic fertilization, and to the spatially heterogeneous development of weeds and disease that are less controlled by crop protection treatments.

Among the difference of treatments between organic and conventional management modes, the fertility management, and in particular the nitrogen fertilization, is probably the one most impacting the crop development (biomass, yield, quality), as nitrogen is often the most demanded and most limiting [2–4] primary nutrient for crop production. The nitrogen fertilizer input in organic farming is generally lower than in conventional ones [5] and is often suboptimal [6].

Organic plant production relies on several principles and rules [7,8] including important ones related to the quantity and type of nitrogen sources allowed in organic agriculture. The total amount of livestock manure applied may not exceed 170 kg of nitrogen per year and hectare of agricultural area used [7]. Other authorized nitrogen sources (guano, soil mineralization, etc.) are not included in these 170 kg of nitrogen; however, those other sources have to be limited according to good agricultural practices [9]. Mineral nitrogen fertilizers shall not be used in organic agriculture.

Additionally, an important part of the nitrogen input in organic farming systems is under organic form and has to be mineralized before it can be assimilated by the crops. The dynamic of the mineralization process depends on several factors, including the soil temperature (season and climate) and microbial populations [10], the type of organic fertilizer, as well as the time when the fertilizers are applied on the field, which is limited [6] and sometimes corresponds in northern climates to the post-harvest period at the end

of summer [11]. This results in a release of nutrients from organic fertilizers that is very often not synchronized with crop uptake, and in a mineralization that can take place at times when no crops are present [12], which may typically lead, in northern climates, to nitrate leaching during a wet winter before crop emergence. Inversely, the mineral nitrogen fertilizers used in conventional farming are readily available to crops and their application timing can be easily tuned and subdivided in several sidedress applications to match the crop need almost perfectly throughout the growing season. Pang and Letey [13] suggested, through a model simulation based on United States data, that due to a bad synchronization of mineralization process and crop needs, it would be difficult to meet the peak nutrient demands of a crop with a very high maximum N-uptake rate, such as maize, by using only organic N, without excessive N in the soil before and after crop growth, and that even a 300 kg ha$^{-1}$ nitrogen input (approximately the double of the maximum amount authorized by EU organic farming regulations) would result in only 70% of the potential maize yield.

Few studies focused on the assessment of the discrimination of organic and conventional crops with remote sensing. Denis [14] showed that multispectral satellite sensors (KOMPSAT-2) enabled full discrimination between 20 organic or organic in conversion and 23 conventional wheat fields, at the soft dough growth stage (Feekes 11.2) as part of an operational farm in Germany, through the use of a simple spectral index (panchromatic/near-infrared), and that a very high discrimination level was achieved from a spatial heterogeneity index. Denis and Tychon [15,16] showed that remote sensing spectral and spatial heterogeneity indices derived from a SPOT 5 satellite image enabled highly significant statistical differences between 50 organic and 50 non-organic cotton fields in Burkina Faso, West Africa, by using univariate and multivariate linear models, with up to 75% discrimination performance. Balashova et al. [17] compared 40 samples equally shared between one organic and one conventional maize parcel in Ohio, United States. Both in situ hyperspectral measurements and satellite images (Landsat-7-ETM+ and Landsat-8-OLI) showed statistically significant differences between the two management modes through single spectral band values or spectral indices, in the visible (VIS), near-infrared (NIR), and short-wavelength infrared (SWIR) spectral ranges. The authors suggested that the observed spectral differences between maize management modes would be due to a more rapid senescence in conventional maize induced by the application of herbicides at maturity aiming to speed up maize maturity as well as to a "*complex combination of factors*". A linear discriminant analysis enabled a level of discrimination (overall classification accuracy) of 100% between organic and conventional samples from satellite derived vegetation indices. However, the very limited dataset used in that study (two fields) considerably limits its scope. Ducati et al. [18] assessed the discrimination level between 46 conventional vineyards and 12 organic ones in the Loire Valley in France with Terra-ASTER (Advanced Spaceborne Thermal Emission and Reflection Radiometer) satellite images. Three statistical linear discriminant analyses using the two visible spectral bands, the seven near-infrared and short-wave infrared spectral bands or all nine spectral bands resulted in 69%, 91%, and 91% classification accuracy, respectively. The normalized difference vegetation index (NDVI) gave no statistically significant difference between management modes. However, the important number of explanatory variables used in the linear discriminant analysis (seven or nine), the small number of observations (58), and the absence of validation may lead to a possible overfitting of the computed model and consequently considerably reduce the significance of the achieved classification accuracy. The authors suggest that the observed spectral differences between vineyard management modes would primarily be due to the chemical treatments used in conventional viticulture through the impact they have on the vine leaf composition and cell structure.

Other studies focused on the remote sensing characterization of maize that received organic or mineral fertilization but that was not, however, officially categorized as organic or conventional. Maresma et al. [19] compared, in a one-year field trial on irrigated maize in Spain, the impact of organic (pig slurry manure, 150 kg N ha$^{-1}$, which is close to the maximum legally authorized nitrogen fertilizer amount of 170 kg ha$^{-1}$ in organic manage-

ment) and mineral (250 kg N ha$^{-1}$, identified as very close to the nitrogen dose enabling the maximum grain yield potential in the study site) fertilizations. Organic fertilized plots presented significantly smaller unmanned aerial vehicle (UAV) vegetation index values for NDVI and Wide Dynamic Range Vegetation Index (WDRVI), Gitelson [20]), smaller (but not significantly) chlorophyll content (measured with a chlorophyll meter), similar crop height, and almost half the grain yield. Yang et al. [21] used an airborne CASI sensor in hyperspectral mode to distinguish, with a decision-tree algorithm, between 80 experimental maize plots amended with manure treatments in a field, and 90 experimental maize plots amended with chemical fertilizers in an adjacent field. They reached a classification accuracy of 96.5%.

Numerous non-remote sensing studies highlighted the effective difference between organic and conventional maize fields in terms of the yield and weed presence. These reinforce the hypothesis that organic and conventional management modes may potentially be discriminated by remote sensing. Crop yield may be considered to be a good indicator to compare the impact of management modes on crops because it integrates the impact of all crop treatments, crop management techniques, and the resulting growing conditions (the presence of weeds and pathogens, soil fertility, etc.) influencing the crop development during the whole growing season. According to four meta-analyses comparing organic and conventional agriculture yields globally [22–25], it appears that when considering mainly developed countries, the ratio of organic maize yield/conventional maize yield ranged from 81% in [25] to 89% (range: 60–141%) in [22] (87% in [23] and in [24] (values computed from the databases accompanying these two papers by excluding one important outlier presenting a ratio of 7.9 for maize in [26])). When focusing on Germany, a study by [27] found a grain yield ratio of 84% (range: 78.6–91.6%). As a comparison, the DOK (bio-Dynamic, bio-Organic and Conventional) Trial in Switzerland reported a ratio of 89–91% for silage maize [28,29]. This tendency in Germany was reinforced by the fact that organic farms are often located in less favorable environments [30]. Studies comparing the weed abundance in organic and conventional maize fields systematically show higher weed abundance in organic fields compared to conventional ones [31–34]. This lower weed abundance in organic crops (not only maize) is explained by the use of herbicides in conventional fields and also by the generally lower crop density in organic fields that allows weeds to develop more abundantly [35,36], personal communication of Doreen Gabriel, Institut für Pflanzenbau und Bodenkunde, Julius Kühn-Institut, Bundesforschungsinstitut für Kulturpflanzen, Germany, 2017]. From a remote sensing point of view, this generally higher abundance of weeds in organic maize fields may counterbalance, at least partially, their generally lower maize biomass. Micskei [37] compared the effect of an amount of farmyard manure and mineral fertilizers equivalent to 132 kg N ha$^{-1}$ on the growth of maize during three years in a long-term maize monoculture field experiments in Hungary. The results showed that mineral fertilization led to a highly significant higher whole foliage nitrogen content and ear leaf chlorophyll content (measured with a chlorophyll meter) on average over the three years, a largely higher maximum plant leaf area, and a delayed senescence of approximately 10 to 20 days (data available for two years only).

Finally, numerous studies demonstrated the ability of remote sensing sensors for the characterization of various maize field/plot/plants properties, such as maize leaf nitrogen or chlorophyll content [38–41], the green Leaf Area Index (LAI) [39,42,43], LAI [44], Plant Area Index (PAI) [45], total biomass [44], plant dry mass [39,46], grain yield [47], soil mineral nitrogen [48], and nitrogen fertilization rates [49–52].

## 2. Materials and Methods

Figure 1 presents the global method applied to in situ and satellite data for the study of the discrimination between organic and conventional maize fields.

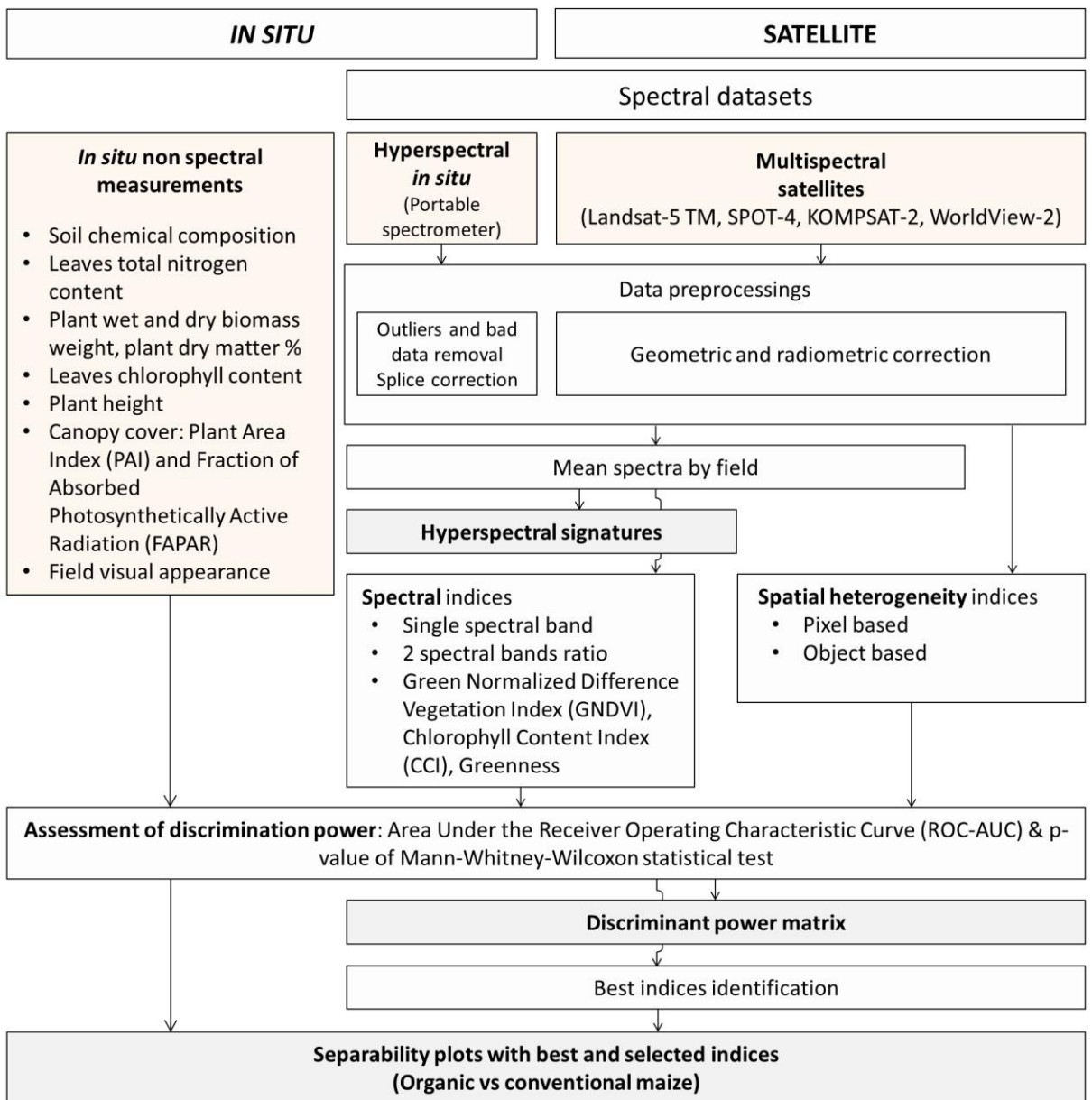

**Figure 1.** Global method applied to in situ and satellite data for the study of the discrimination between organic and conventional maize fields.

### 2.1. Study Area and Studied Fields

The study area (Figure 2) was located in central eastern Germany, near Leipzig city, covered 15 × 20 km, and was flat. The Koeppen climate class is Cfb Oceanic, warm maritime temperate climate, fully humid, warm summer [53], with minimum–maximum annual temperature (monthly average) of −1 °C to +18 °C and average monthly precipitation of 42 mm.

Depending on the indices computed, between 24 to 32 maize fields (Figure 2) were involved in the analysis, of which approximately half were organic and half were conventional. The mean area of the studied fields was around 22 ha for conventional maize (range of 4 ha to 83 ha) and 34 ha for organic maize (range of 4 ha to 98 ha). All fields were managed by a single company applying very similar treatments for a given management mode.

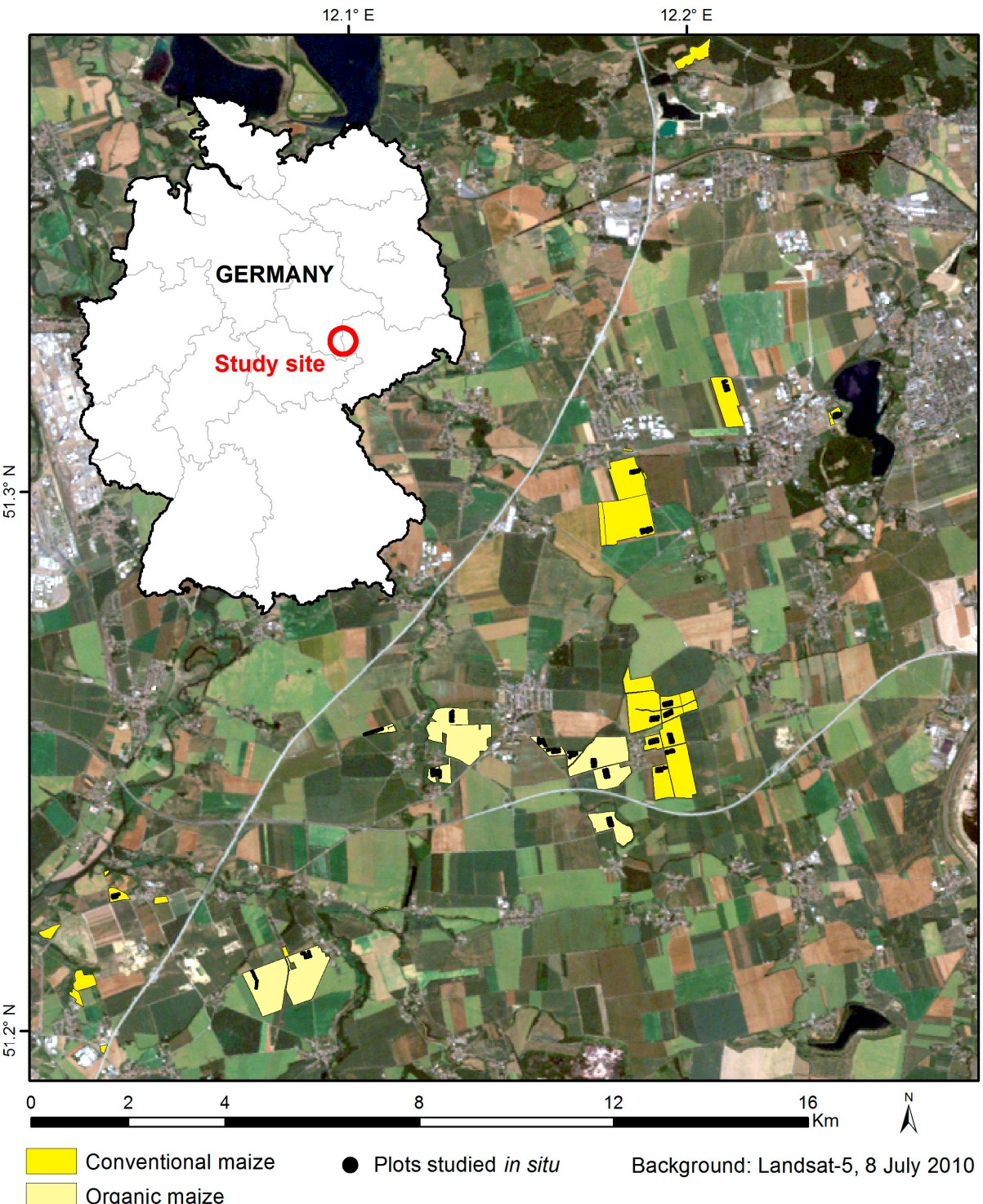

**Figure 2.** Localization of the study area and studied organic and conventional maize fields and plots.

*2.2. In Situ Indices*

2.2.1. Field Survey and Field Sampling

In situ measurements were carried out on 12 organic and 12 conventional maize fields during a field survey that occurred between the 6th and 10th of August 2010. The maize growth stage was around reproductive 3, i.e., yellow kernels on the outside with a milky inner fluid (Figure 5). The sampling sites were always away from the field margins to avoid the field margin effect on plants (systematic lower vitality at the field margins). The

10 sampling plots among a field were spaced by 30 m from each other and distributed on two parallel lines.

### 2.2.2. Soil Chemical Composition

One soil sample of around 300 g was collected by field in a 5 m × 5 m square. This sample was composed of 20 subsamples evenly distributed in the sampling square. Each subsample was collected with a thin auger of a diameter of two centimeters to a depth of five centimeters. The soil samples were immediately pouched in a hermetic plastic bag and kept at 4 °C until laboratory analysis three to four days later.

Key soil parameters were assessed: pH $H_2O$, pH KCl0, and pH KCl1, total nitrogen (ammonium, nitrate, nitrite, and organic nitrogen) with the Kjeldahl method [54], organic carbon and humus with the Walkley–Black method [55], minerals (potassium (K), sodium (Na), magnesium (Mg), and calcium (Ca)) by atomic absorption spectrometry, and phosphorus (P) by colorimetry.

### 2.2.3. Leaves Total Nitrogen and Dry Matter

One leaf sample was collected per field in a 5 m by 5 m square. The leaf sample was composed of 20 leaves selected at the top of the canopy; however, the very last leaves were avoided due to their small development and their low contribution to global canopy reflectance. Leaf samples were immediately pouched in a hermetic plastic bag and kept at 4 °C until laboratory analysis three to four days later.

The total nitrogen content was estimated with the Kjeldahl method [54] on the samples dried at 70 °C. The leaf dry matter was estimated according to Equation (1).

$$\text{Dry matter at 70 °C (\%)} = \frac{\text{Dry weight at 70 °C (g)}}{\text{Wet weight (g)}} * 100. \tag{1}$$

### 2.2.4. Plant Wet and Dry Biomass Weight, Dry Matter

One plant sample was collected per field. It consisted of one row three meters long of maize entire plants. The wet biomass weight was measured with a portable field scale with an accuracy of 100 g directly in the field.

A three-plant subsample was extracted from each main sample and pouched in a plastic bag until it was weighted two to three days later with an accuracy of 1 g as wet biomass. These subsamples were then dried at 105 °C, and the dry weights were estimated with an accuracy of 1 g. The entire plant dry matter was computed according to Equation (2).

$$\text{Dry matter at 105 °C (\%)} = \frac{\text{Dry weight at 105 °C (g)}}{\text{Wet weight (g)}} \times 100. \tag{2}$$

### 2.2.5. Leaf Chlorophyll Content

Five chlorophyll content measures were realized in each of the 10 plots by field, i.e., 50 measures per field and 1200 measures in total. Each measure was spaced by four meters in a maize row, on leaves selected at the top of the canopy; however, the very last leaves were avoided due to their small development and their low contribution to the global canopy reflectance.

The chlorophyll content was estimated with a Chlorophyll Content Meter CCM-200 from Opti-Sciences. The CCM-200 is a hand-held instrument designed for the rapid, nondestructive determination of the chlorophyll content in intact leaf samples (Figure 3a). The CCM-200 uses transmittance in two wavelengths. One falls within the chlorophyll absorbance range (red at 653 nm) while the other serves to compensate for mechanical differences, such as tissue thickness (near infrared at 931 nm). The Chlorophyll Content

Index (CCI) is computed according to Equation (3) [56]. The CCI spatial heterogeneity was estimated by the standard deviation of all CCI values by field.

$$\text{Chlorophyll Content Index (CCI)} = \frac{\text{Transmittance at 931 nm}}{\text{Transmittance at 653 nm}}. \tag{3}$$

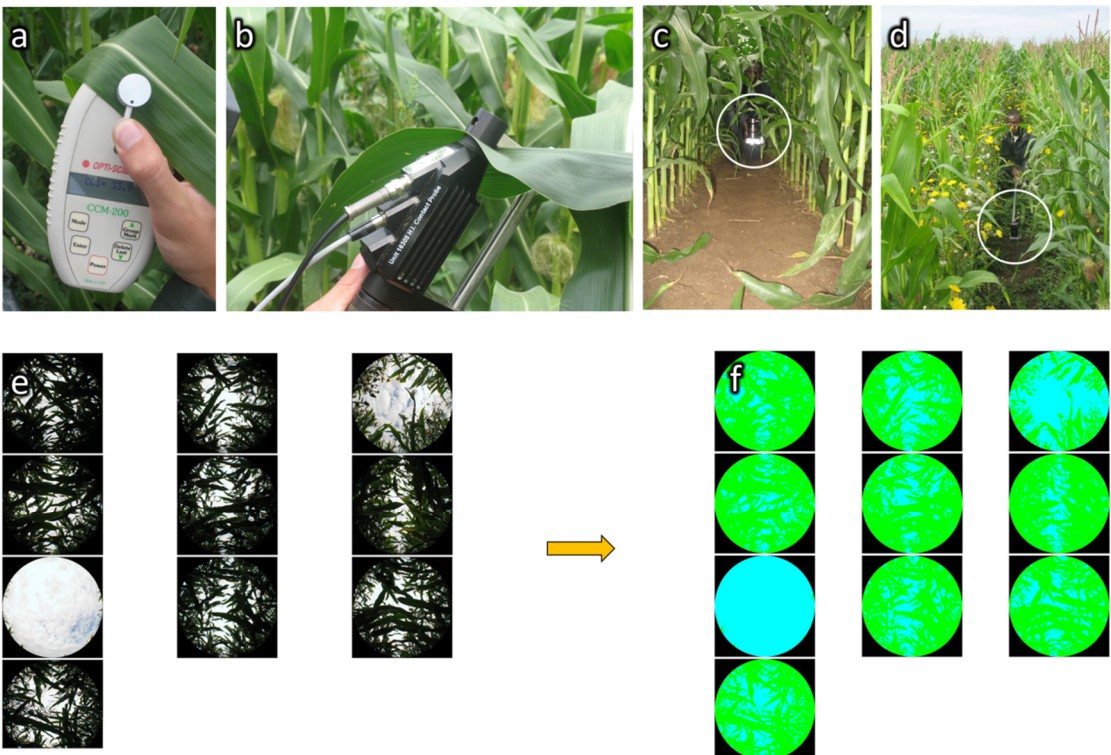

**Figure 3.** Illustrations of three types of in situ parameter measurements. (**a**) Chlorophyll content measurement on a conventional maize leaf with a Chlorophyll Content Meter CCM200; (**b**) Hyperspectral reflectance measurement on a conventional maize leaf with an high intensity contact probe of an Analytical Spectral Device (ASD) FieldSpec3 spectroradiometer equipped with a leaf clip; (c-d-e-f) Hemispherical pictures for the Plant Area Index (PAI) and Fraction of Absorbed Photosynthetically Active Radiation (FAPAR) measurements in conventional (**c**) and organic (**d**) maize fields and sample of corresponding raw (**e**) and classified (**f**) pictures for an organic maize field.

### 2.2.6. Plant Height

Five plant height measures were realized in each of the 10 plots by field, on five neighboring plants, i.e., 50 measures per field and 1200 measures in total. Plant height was measured with a two-meter-long folding rule. The plant height spatial heterogeneity was estimated by the standard deviation of all plant height values by field.

### 2.2.7. Canopy Cover (PAI, FAPAR)

Eleven canopy cover measures were realized by field, i.e., 264 measures in total. Canopy cover was estimated through the analysis of vertical hemispherical pictures taken with a Besel Super Wide Fisheye lens 0.25 ×W-52025 equipped with a Macro and placed on a camera Canon PowerShot A590 IS. The camera was fixed on a two meters long boom and placed, pointing up, at the base of the plants in the middle of two maize rows (Figure 3c,d). Pictures were taken in manual mode with the camera self-timer set to 10 s. Focal and aperture speed were adjusted depending on the illumination conditions to clearly enable the differentiation between plants (maize and weeds) and the sky.

CAN-EYE software [57] was used to classify the hemispherical pictures in two classes: plant and sky (Figure 3e,f). An approximation of the two following biophysical variables was derived from CAN-EYE: (i) PAI: one sided plant area per unit horizontal ground

surface area adapted from the LAI definition of [58] cited in [57]; (ii) the Fraction of Absorbed Photosynthetically Active Radiation (FAPAR): the fraction of the incoming solar radiation in the Photosynthetically Active Radiation spectral region that is absorbed by a photosynthetic organism.

### 2.2.8. Hyperspectral Leaf Reflectance

Five reflectance measurements were realized in each of the 10 plots by field, i.e., 50 measures per field and 1200 measures in total. Each measure was spaced by four meters in a maize row, on leaves selected at the top of the canopy; however, the very last leaves were avoided due to their small development and their low contribution to the global canopy reflectance.

The field hyperspectral reflectance measurements were realized with an Analytical Spectral Device (ASD) FieldSpec3 spectroradiometer in reflectance mode. This sensor records the reflectance in the range 350–2500 nm, with, depending on the wavelength, a spectral resolution (Full-Width-Half-Maximum, FWHM) of 3–10 nm and a sampling interval of 1.4–2 nm [59], and delivers finally, after resampling (personal communication of Gary Fager, Quality control and technical support manager at ASD Inc. (Boulder, CO, USA)), 2151 spectral bands, one per nanometer. An ASD high intensity contact probe equipped with a leaf clip was used to realize measurements on individual leaves (Figure 3b). Optimization was realized at the beginning of each field and when necessary during the field measurements. White reference calibration, necessary to calibrate the spectroradiometer to create a proper reflectance spectrum, was realized before the measurement at each plot.

Spectra preprocessing consisted of: (i) identification and removal of outliers and bad spectra by visualization of each spectra; (ii) a splice correction that adjusted the reflectance values of the three ASD sensors by computing bias values for the visible and near-infrared (VNIR) spectrometer (350–1000 nm) and the short-wave infrared (SWIR) 2 spectrometer (1830–2500 nm) compared to the SWIR 1 spectrometer (1000–1830 nm) and then offsetting them to match the SWIR1 at the splice points [59,60].

The mean reflectance spectra by field were computed. Then, three types of spectral indices were studied: (i) all single spectral band reflectances; (ii) all possible combinations of two spectral bands in the form of a simple ratio ($B_i/B_j$), including the CCI used by the Chlorophyll Content Meter CCM-200 [61] (Equation 4) and the Greenness index, a chlorophyll related index [62] (Equation (5)); and (iii) the Green Normalized Difference Vegetation Index (GNDVI), a chlorophyll related index [63] (Equation (6)).

$$\text{Chlorophyll Content Index (CCI)} = \frac{\text{NIR (931 nm)}}{\text{Red (653 nm)}} \tag{4}$$

$$\text{Greenness index (G)} = \frac{\text{Green}}{\text{Red}} \tag{5}$$

$$\text{Green Normalized Difference Vegetation Index (GNDVI)} = \frac{\text{NIR} - \text{Green}}{\text{NIR} + \text{Green}}. \tag{6}$$

### 2.3. Satellite Indices

#### 2.3.1. Satellite Imagery Description

Four multispectral satellite images with various spatial and spectral resolutions were acquired at three maize growth stages: one Landsat-5-TM image at growth stage reproductive 1, one WorldView-2 image at growth stage reproductive 3 and one KOMPSAT-2 and one SPOT-4 image at the maturity growth stage (Table 1).

#### 2.3.2. Satellite Images Preprocessing

Landsat-5 images were acquired in preprocessing level 1T, SPOT-4 in level 2A, KOMPSAT-2 in level L1G, and WorldView-2 in level ORStandard2A. These preprocessings deliver geometrically corrected images in which the pixel values are proportional to the radiometrically corrected radiance. An additional georeferencing was done by using a series of

ground control points extracted from a vectorial file of the field delineations as reference. The geometric correction was validated visually by checking the matching between the images and the field delineations vectorial file.

**Table 1.** Satellites and sensor features used in this study, acquisition date, and corresponding maize growth stage and number of maize fields studied by image and crop management mode. Swath: strip of the Earth's surface viewed by a sensor; MS: multispectral; PAN: panchromatic; M: monospectral; VIS: visible; NIR: near-infrared; TIR: thermal infrared; SWIR: short-wave infrared.

| Satellite Name | Sensor Name | Swath (km) | Spectral Bands | Spatial Resolution (m) | Acquisition Date | Maize Growth Stage | Number of Fields Studied | |
|---|---|---|---|---|---|---|---|---|
| | | | | | | | Conventional | Organic |
| Landsat-5 | Thematic Mapper (TM) | 185 | 7 MS (blue, green, red, NIR, SWIR, TIR, SWIR) | 30 (TIR: 120) | 8 July 2010 | Reproductive 1 | 20 | 12 |
| WorldView-2 | MS: WV110 PAN: WV60 | 16.4 | 8 MS (coastal, blue, green, yellow, red, red-edge, NIR, NIR2) PAN (450–800 nm) | MS: 2 PAN: 0.5 | 10 August 2010 | Reproductive 3 | 14 | 9 |
| KOMPSAT-2 | Multispectral Camera (MSC) | 15 | 4 MS (blue, green, red, NIR) PAN (500–900 nm) | MS: 4 PAN: 1 | 21 September 2010 | Maturity | 17 | 11 |
| SPOT-4 | High-Resolution Visible and InfraRed (HRVIR) | 60 | 4 MS (green, red, NIR, SWIR) 1 M (610–680 nm) | MS: 20 M: 10 | 24 September 2010 | Maturity | 13 | 12 |

### 2.3.3. Extraction of Satellite Images Pixel Values by Field

The extraction of satellite image pixel values by field was realized using field delineations that were adapted to each image and each field to avoid mixed pixels of field margins and the impact of external elements, such as high trees and their shadow at the field margins for example. Depending on the image extent, localization, spatial resolution, and on the presence of clouds, the number of fields for which crop pixel values could be extracted varied from one image to another (Table 1).

### 2.3.4. Satellite Indices Computation

Two types of satellite indices were computed: spectral indices and spatial heterogeneity indices.

### Spectral Indices

Spectral indices correspond to: (i) the mean pixel values by field of each spectral band of satellite images; (ii) all possible combinations of two of these values in the form of a simple ratio ($B_i/B_j$) among which the Greenness index (G) and an approximation of the CCI; (iii) the GNDVI (Equations (4)–(6)).

### Spatial Heterogeneity Indices

The spatial heterogeneity of the fields was assessed through pixel-based and object-based indices. Each index is defined in Table 2. All spatial heterogeneity indices were computed using eCognition Developer software.

The pixel-based spatial heterogeneity was calculated by considering the value of all individual pixels in a given field and corresponds to the standard deviation of the pixel values by field and to various Haralick textural parameters [64–66] derived from the Gray-Level Co-occurrence Matrix (GLCM) and the Gray-Level Difference Vector (GLDV). The GLCM is a frequency tabulation of how often different combinations of two neighbor pixel gray levels occur in an image.

A different co-occurrence matrix exists for each direction (vertical, oblique, horizontal) chosen to establish the neighborhood spatial relationship between pixels. To receive directional invariance, the GLCM of all four directions (0°, 45°, 90°, 135°) are summed. The GLCM are normalized. Adapted from [67]. The GLDV is the sum of the diagonals of the GLCM [67]. The standard deviation was computed for each spectral band and the GLCM and GLDV indices were computed from a mean image made from all spectral bands of a given image.

**Table 2.** Definition of the field spatial heterogeneity indices computed from the satellite images in this study adapted from [68]. GLCM: Gray-Level Co-occurrence Matrix; GLDV: Gray-Level Difference Vector.

| Pixel/Object-Based | Name | Definition |
|---|---|---|
| PIXEL | Standard deviation | Standard deviation of pixel values by field. |
| | GLCM standard deviation | Standard deviation of the GLCM values. |
| | GLCM homogeneity | Measure of the local homogeneity in the image. Homogeneity is high if higher values concentrates along the GLCM diagonal. |
| | GLCM contrast | Contrast is the opposite of homogeneity. Measure of the amount of local variation in the image. |
| | GLCM Angular 2nd moment | Measure of the local homogeneity. The value is high if some elements of the GLCM are large and the remaining ones are small. |
| | GLCM entropy | The value is high if the elements of the GLCM are distributed equally. It is low if the elements are close to either 0 or 1. |
| | GLCM dissimilarity | Similar to contrast. High if the local region has a high contrast. |
| | GLDV Angular 2nd moment | Measure of the local homogeneity. The value is high if some elements are large and the remaining ones are small. |
| | GLDV entropy | The values are high if all elements have similar values. It is the opposite of GLDV Angular Second Moment. |
| OBJECT | Mean of densities of sub-objects | Mean value of the densities of the sub-objects. The density index describes the distribution in space of the pixels of an image object. The densest shape is a square; the more an object is shaped like a filament, the lower its density. |
| | Standard deviation of densities of sub-objects | Standard deviation calculated from the densities of the sub-objects (confer previous definition). |
| | Mean of asymmetries of sub-objects | Mean value of the asymmetries of the sub-objects. The asymmetry index describes the relative length of an image object. It corresponds to the ratio of the lengths of the major and minor axes of an ellipse approximated around a given image object. The index value increases with this asymmetry. Similar to the length/width ratio of an image object. |
| | Standard deviation of asymmetries of sub-objects | Standard deviation of the asymmetries of the sub-objects (confer previous definition). |
| | Standard deviation of mean values of sub-objects | Standard deviation of the mean values of the sub-objects. This index might appear very similar to the simple standard deviation computed from the single pixel values; however, it can be more meaningful because—assuming an adequate segmentation—the standard deviation is computed over homogeneous and meaningful areas. |
| | Average mean difference of neighbor sub-objects | The contrast inside an image object expressed by the average of all mean absolute difference of each sub-object with its adjacent sub-objects of the same object. |

The object-based spatial heterogeneity was calculated by considering the value of groups of spectrally homogeneous neighbor pixels, called spectral objects and obtained by a preliminary segmentation of the image. This method enables the computation of heterogeneity indices by using, for example, the properties of an object, such as its shape, area, or texture, or by using the relative spatial organization and values of these objects in one or several object levels. Two object levels were used, the upper one corresponding to the field delimitations and the lower one to smaller spectral objects inside the fields (field

sub-objects). Density and asymmetry indices relate to the shape of the objects and are, thus, not computed from a particular spectral band. The two indices mean of densities of sub-objects and mean of asymmetries of sub-objects were computed from a single spectral band: the panchromatic band for KOMPSAT-2, WorldView-2, and SPOT-4, and the blue-green band for LANDSAT-5.

### 2.4. Statistical Assessment of the Discriminating Power of Indices

The level of separation between conventional and organic maize was assessed through the Area Under the Receiver Operating Characteristic Curve (ROC-AUC). The ROC-AUC is a common method to assess classifier performance [68–70] to predict a binary outcome [71].

The ROC-AUC values vary between 0 and 1. A ROC-AUC of 0 or 1 corresponds to a perfect separation between the two groups, with no overlapping points. A ROC-AUC of 0.5 corresponds to a random classification or to a classification that could be achieved by an uninformative classifier, i.e., a perfect mix between the two groups. In this study, the ROC-AUC values are smaller/higher than 0.5 when conventional fields present generally higher/lower values than organic fields, respectively. The ROC-AUC does not reflect the magnitude of the difference between the two groups.

The ROC-AUC presents two main advantages. First, ROC-AUC is insensitive to the number of observations (fields) compared. This property enables use of it to compare classifier performance over situations presenting significantly different number of observations. Secondly, ROC-AUC is insensitive to changes in the class distribution [68,72], i.e., changes in the relative number of observations in each of the two classes. This property enables use of it to assess the separability among differently imbalanced datasets (i.e., presenting different number of observations in the two groups), as those encountered in this study presenting imbalances varying between 48%/52% (rather balanced) up to 38%/63% (rather imbalanced).

The ROC-AUC was computed in R software with the ROCR package [73,74] via the prediction and performance functions.

The statistical significance of the difference between organic and conventional fields achieved with the computed indices was assessed through the *p*-value of the Mann–Whitney–Wilcoxon (MWW) non-parametric statistical hypothesis test [75,76]. This test was selected because this study deals with comparisons of independent samples from two populations of which the normality of distribution was not always met. As a rank test [77,78], this is less affected by extreme values than tests based on observed numerical values. For a given dataset with a given number of observations, the MWW test statistic W is equivalent to the ROC-AUC [79–81]; however, this relation breaks when considering datasets of different sizes as the MWW W statistic depends on the sample size while ROC-AUC does not. The test was carried out in R software through the wilcox.test function, as a two-sided test.

### 3. Results

#### 3.1. In Situ Results

3.1.1. Soil Chemical Composition

The chemical composition of soil samples of maize fields and their related discriminating power are presented in Figure 4. Only potassium and sodium (Figure 4h,i) presented statistically significant differences between the management modes (*p*-values in the range (0.01–0.05)) with a moderate discrimination level (ROC-AUC of 0.25 and 0.23) and higher values for conventional fields. However, the mean and median values were slightly higher in conventional maize fields for all soil nutrient parameters, except for phosphorous (Figure 4j) and for the median value of calcium (Figure 4g).

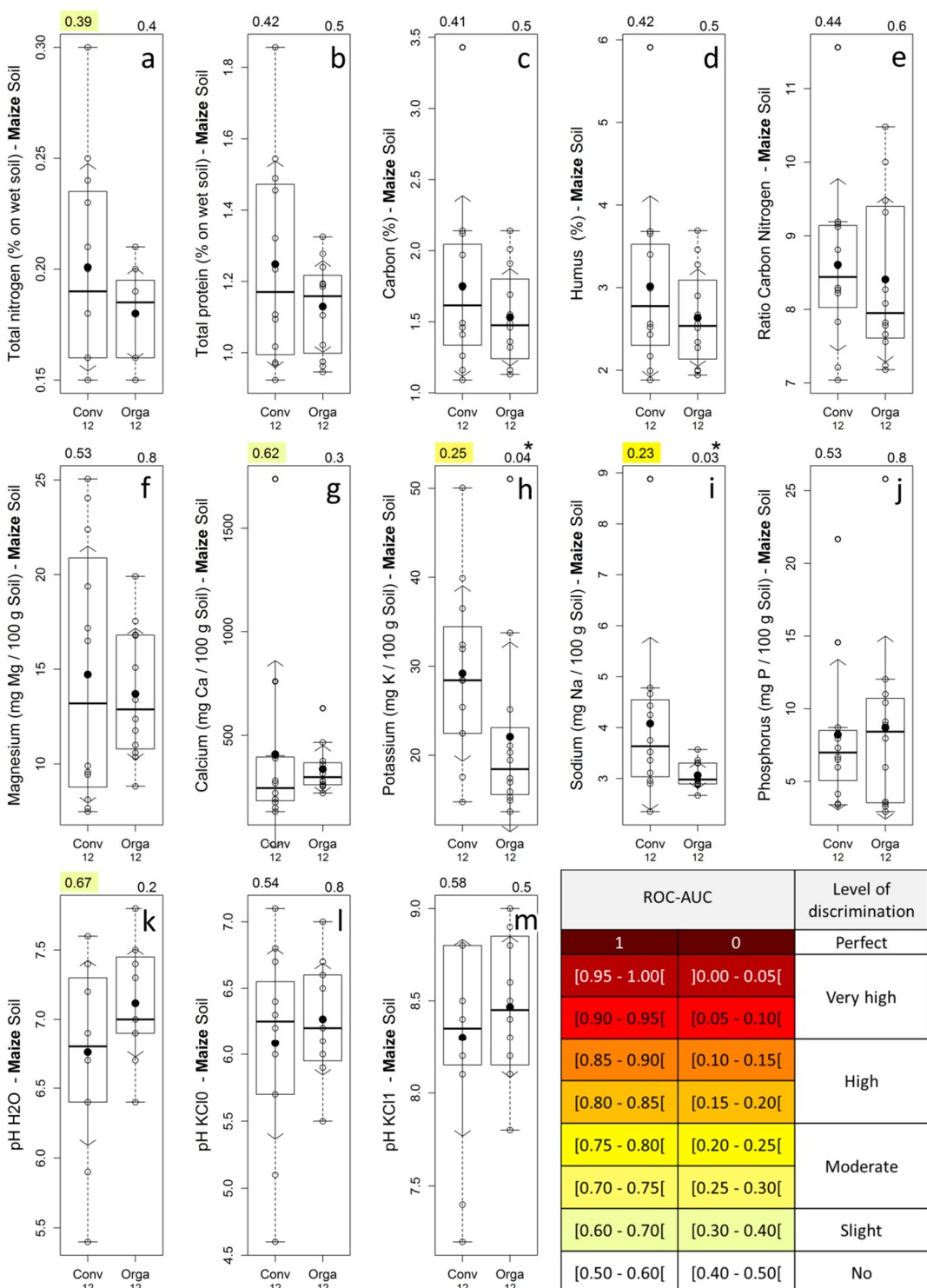

**Figure 4.** Soil samples chemical composition for the 12 conventional and 12 organic maize fields and related discrimination power expressed as the ROC-AUC (**a–m**). Box-and-whisker plot: empty circles: observed values; black filled circles: mean value; arrows: standard deviation; box: InterQuartile Range (IQR) delimited by the 1st (Q1) and 3rd (Q3) quartiles, with the median indicated by a horizontal line; whiskers: the most extreme data points that are at a distance of less than 1.5 times the IQR from the edges of the IQR box (i.e., Q1 − 1.5 × IQR and Q3 + 1.5 × IQR); top-left corner: ROC-AUC (Area Under the Receiver Operating Characteristic Curve) value with a colored background referring to ROC-AUC color scale; top-right corner: *p*-value of the MWW statistical test, with stars representing its level of statistical significance: No star= not significant, * = *p*-value ∈[0.05–0.01[, ** = *p*-value ∈[0.01–0.001[, *** = *p*-value ≤ 0.001; Conv/Orga: Conventional/Organic; numbers below x labels: number of fields belonging to each management mode.

### 3.1.2. Visual Appearance of Maize Fields

The visual appearance of fields for typical organic and conventional fields is illustrated in Figure 5. Conventional maize presents a darker green and a more closed canopy compared to organic fields. Weeds were very abundant is some organic fields (sunflowers, poppies, and chamomile, mainly) while systematically absent in conventional ones due to the use of herbicides.

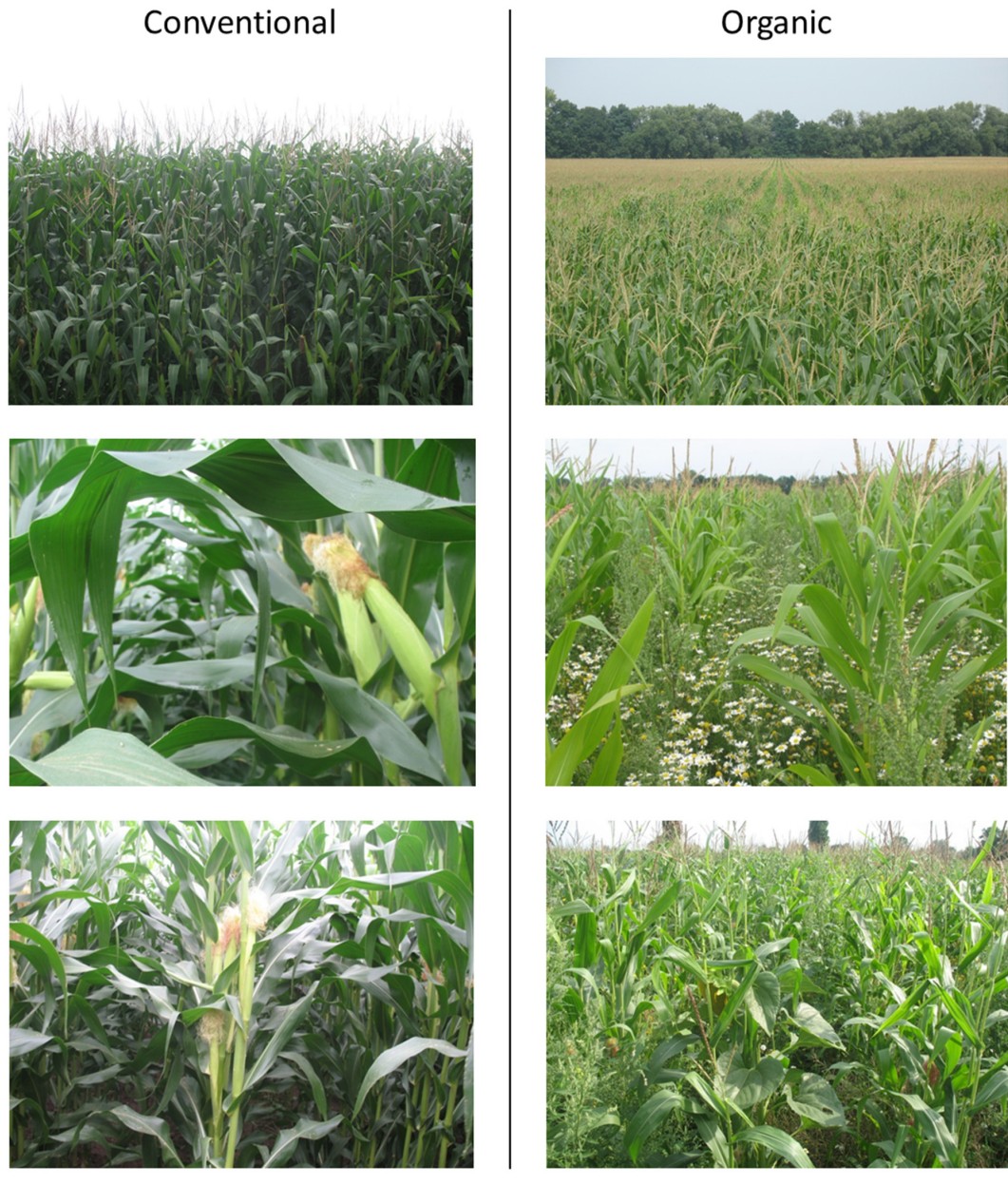

**Figure 5.** Visual appearance of some typical conventional (left) and organic (right) maize fields.

### 3.1.3. Leaf Total Nitrogen Content and Leaf Dry Matter Percentage

The total nitrogen content of dry maize leaves was very significantly (*p*-values < 0.0001) higher for conventional fields, and the total nitrogen content of wet maize leaves was slightly less significantly (*p*-values of approximately 0.001) higher for conventional fields (Figure 6a,b). Nearly complete separation between the two management modes was observed for dry leaves (except for one field) and a high discrimination level was observed for wet leaves. Maize leaves dry matter was significantly (*p*-value = 0.02) higher for organic fields and provided a moderate discrimination level (Figure 6c).

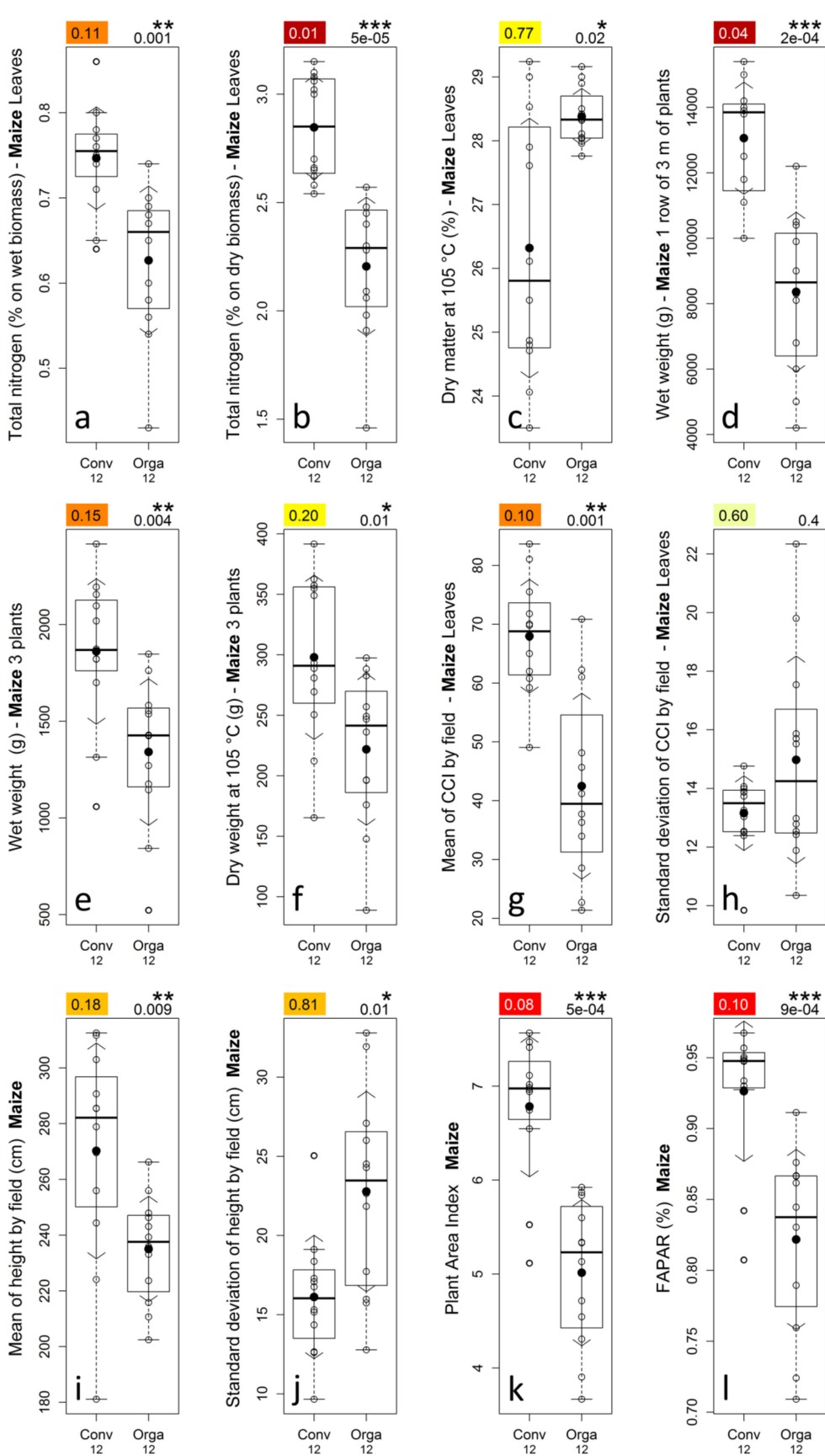

**Figure 6.** Parameters studied in situ for the 12 conventional and the 12 organic maize fields and related discrimination power expressed as the ROC-AUC (top-left value) (**a**–**l**). CCI: Chlorophyll Content Index, and FAPAR: Fraction of Absorbed Photosynthetically Active Radiation. Detailed description of the plots available in Figure 4.

### 3.1.4. Plants Wet and Dry Biomass Weight, Dry Matter Percentage

The wet and dry weights of the maize three-plant samples (Figure 6e,f) were very significantly (*p*-value of 0.004 and 0.01 respectively) higher for conventional fields and provided high and moderate discrimination levels, respectively, while the dry matter (plot not shown) was not significantly (*p*-value of 0.2) higher for organic fields.

The wet weight of the maize three-meter-long plant row (Figure 6d) was very significantly (*p*-value < 0.001) higher for conventional fields with a nearly complete separation between conventional and organic fields. This last weight parameter, due to the larger sample size (row of three meters of plants), is more representative than those measured on three maize plants.

### 3.1.5. Leaves Chlorophyll Content

The maize mean chlorophyll content by field (Figure 6g) was very significantly (*p*-value = 0.001) higher for conventional fields and provided a high discrimination level. Nearly complete separation (except for four fields) was observed between conventional organic fields.

The maize standard deviation of chlorophyll content by field (Figure 6h) was not significantly (*p*-value = 0.4) different between conventional and organic fields. However, some organic maize fields presented very high standard deviations.

### 3.1.6. Plants Height

The maize mean height by field (Figure 6i) was very significantly (*p*-value = 0.009) higher for conventional fields and provided a high discrimination level. The maize standard deviation of height by field (Figure 6j) was very significantly (*p*-value = 0.01) higher for organic fields and provided a high discrimination level.

From this analysis, organic maize fields can be characterized as generally shorter and presenting a more heterogeneous canopy compared with conventional fields.

### 3.1.7. Canopy Cover (PAI, FAPAR)

The maize mean PAI and FAPAR values by field (Figure 6k,l) were very significantly (*p*-value < 0.001) higher for conventional fields. Nearly complete separation (except for two fields) was observed between conventional and organic fields.

### 3.1.8. Leaves Hyperspectral Reflectance

Reflectance in the 500–650 nm part of the VIS range, and especially in the green (around 550 nm) was clearly higher for organic maize fields compared to conventional ones (Figure 7a). Visible reflectance below 500 nm and over 650 nm showed constant values over the management modes. Higher peak reflectance in the green corresponded visually to a lighter green while lower peak reflectance in the green corresponded to a darker green. This reflectance behavior was associated with the fact that reflectance in that range is influenced by the plant pigments' nature and concentration, including chlorophyll. Chlorophyll pigments are more present in conventional crops, as already observed with the CCM-200 chlorophyll content measurements, which results in a stronger absorption of the green light and, consequently, in darker green canopies/leaves.

For both single spectral band indices and two spectral band ratio indices, the spectral region between 500 and 750 nm presented the best discrimination power (ROC-AUC values) (Figure 7b). Single spectral band indices presented higher discrimination power around 550 nm (green) and 725 nm (red-edge) (very high and high discrimination levels, respectively) with the most discriminant spectral band identified at 570 nm (Figure 7c). Some two spectral band ratio indices presented even higher discrimination power with a few indices presenting up to perfect discrimination (Figure 7b), among which the best index was the ratio 640 nm/690 nm (Figure 7d).

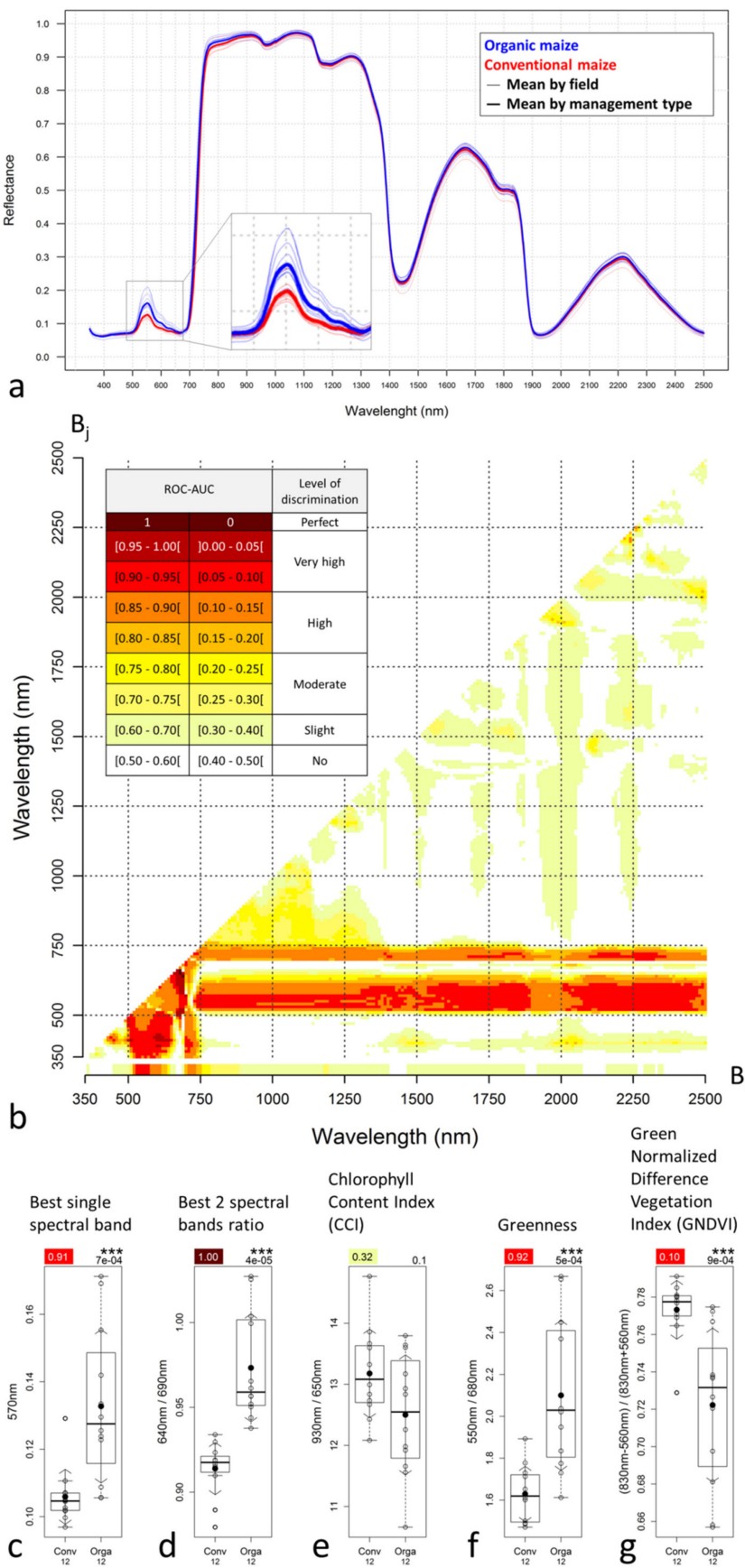

**Figure 7.** Leaf hyperspectral in situ parameters for the 12 conventional and 12 organic maize fields. (**a**) Leaf hyperspectral signatures. (**b**) The level of discrimination between conventional and organic

maize computed from maize leaves hyperspectral measurements and expressed as the Area Under the Receiver Operating Characteristic Curve (ROC-AUC) values. In the matrix, the ROC-AUC values of all possible combinations of two spectral bands under the form of a simple ratio ($B_i/B_j$). In the line below the matrix, the ROC-AUC values of the single spectral bands. The sampling interval of the hyperspectral signal was 10 nm. (**c**–**g**) Leaf hyperspectral reflectance indices and related discrimination power expressed as the ROC-AUC (plot top-left value). Detailed description of the plots available in Figure 4.

Reflectance in the NIR-SWIR (750–2500 nm) range showed very little to no difference between management modes (Figure 7a), which is likely related to the fact that the NIR reflectance range is mainly sensitive to the canopy structure and density and that maize spectra correspond to small pieces of leaves measured in the ASD contact probe, which are consequently not impacted by the canopy structure variation. The NIR spectral region (750–1300 nm) provided moderate discrimination power, while the region after 1300 nm provided very rare discriminant indices (Figure 7b).

The CCI and GNDVI were higher for conventional maize while the Greenness was higher for organic (Figure 7e–g), as expected from the spectral signature analysis. This also confirms the study hypothesis that conventional fields present higher chlorophyll contents. Greenness performed similarly to GNDVI with a very high discrimination level, while CCI, despite visible differences on the box-and-whisker plot, was not significantly different between management modes.

### 3.2. Satellite Results

Figure 8 presents the discrimination power of the different types of indices computed from the four multispectral satellite images, i.e., single spectral band, two spectral bands ratio, CCI, Greenness, GNDVI, and pixel and object-based spatial heterogeneity. Figure 9 presents the box-and-whisker plots for the best indices of each category.

### 3.2.1. Spectral Indices

Globally, some multispectral satellites indices were revealed to be very efficient in crop management mode discrimination for each of the four situations studied with complete or close to complete separation achieved (Figures 8 and 9).

Regarding the best single spectral band indices (Figure 9, first column), the near-infrared band was identified as the most discriminant in all situations, presenting always higher values for the conventional management mode and enabling very highly significant discrimination in all four situations, with a complete separation observed in one situation and a complete separation with the exception of one or two fields in the three other situations. As the reflectance of the NIR spectral band was positively correlated with crop biomass [82], these observations confirm the study hypotheses that organic maize fields present lower crop biomass development compared with conventional fields and that this can be observed by remote sensing.

This analysis is valid for crop growth stages presenting green vegetation but should be nuanced for crop growth stages potentially presenting senescent vegetation as higher NIR reflectance might also be due to earlier senescence. The visual analysis of the two images concerned by a mature growth stage, i.e., the KOMPSAT-2 and SPOT-4 (September), did not enable any conclusions regarding the potential level of senescence of the fields at that time, given that the true color composition of the KOMPSAT-2 image presented colors difficult to interpret and that the spectral bands of SPOT-4 image did not enable a true color composition.

The analysis of the single band part of Figure 8 confirmed the higher discrimination power of the NIR bands compared to the others, with, for some situations, for example for Landsat-5, a very important difference of the discrimination power. These observations demonstrate the very high efficiency of the NIR spectral region in the discrimination of the crop management modes. Depending on the situation, some other spectral bands

(red-edge, red, green, and panchromatic) enable high or very high discrimination. The purple, blue, and thermal infrared (TIR) bands, when considered to be single band, were revealed to be inefficient in all situations where they were present.

Regarding the best spectral band ratio indices (Figure 9, second column), they contained three times out of four an NIR band, combined with various other spectral bands (thermal infrared, purple, and green) depending on the situation. The analysis of the two-band ratio part of Figure 8 revealed that the ratios containing an NIR band were among the most discriminant. The best spectral band ratio (Figure 9, second column) indices enabled complete separation in two situations and a nearly complete separation (except for one field) in the two other situations. The best two spectral bands ratio indices were never meaningfully more discriminant than the best single band indices, except for the KOMPSAT-2 image, for which the best two spectral bands ratio index provided a separation of clearly higher magnitude (Figure 9, second column).

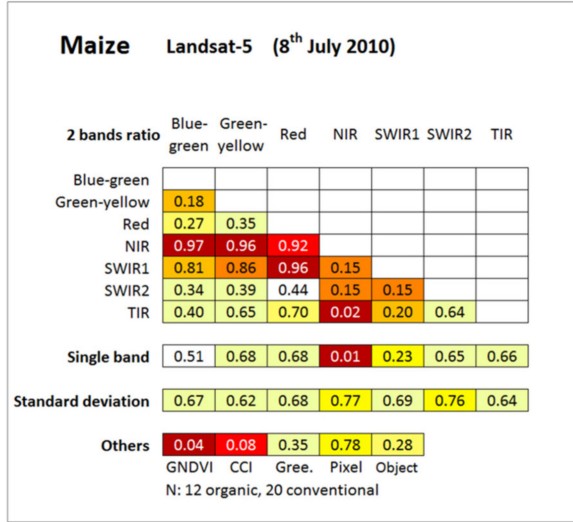

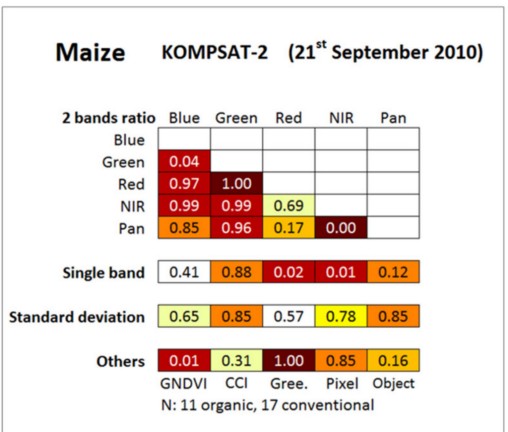

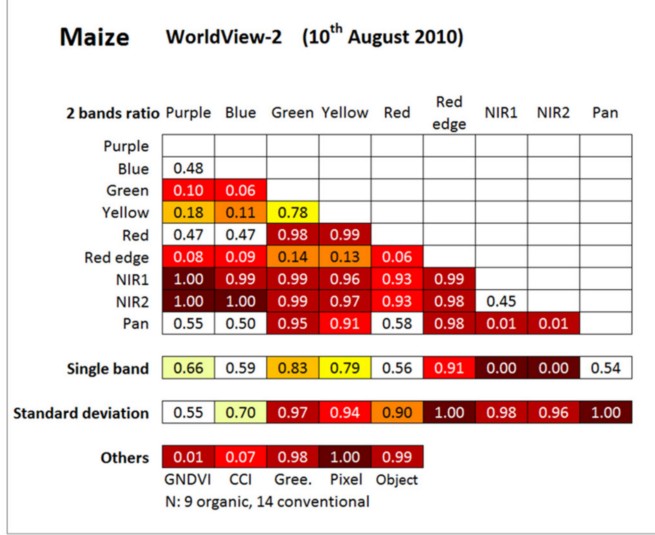

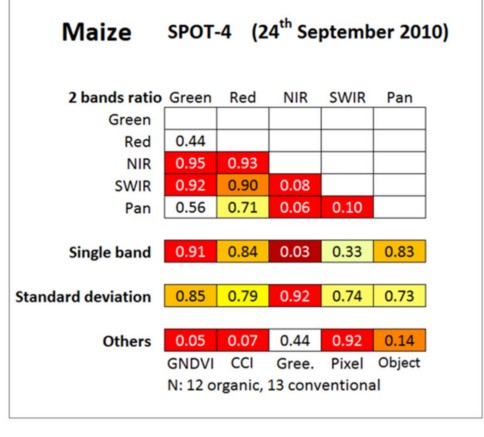

**Figure 8.** Level of discrimination between conventional and organic maize fields computed from the four multispectral satellites images and expressed as ROC-AUC values. GNDVI: Green Normalized Difference Vegetation Index; CCI: Chlorophyll Content Index; Gree = Greenness index; Pixel/Object: best pixel/object-based spatial heterogeneity index; N = number of fields. NIR: near-infrared; SWIR: short-wavelength infrared; TIR: thermal infrared; Pan: panchromatic. Color scale referring to the ROC-AUC (Area Under the Receiver Operating Characteristic Curve) color scale of Figure 4.

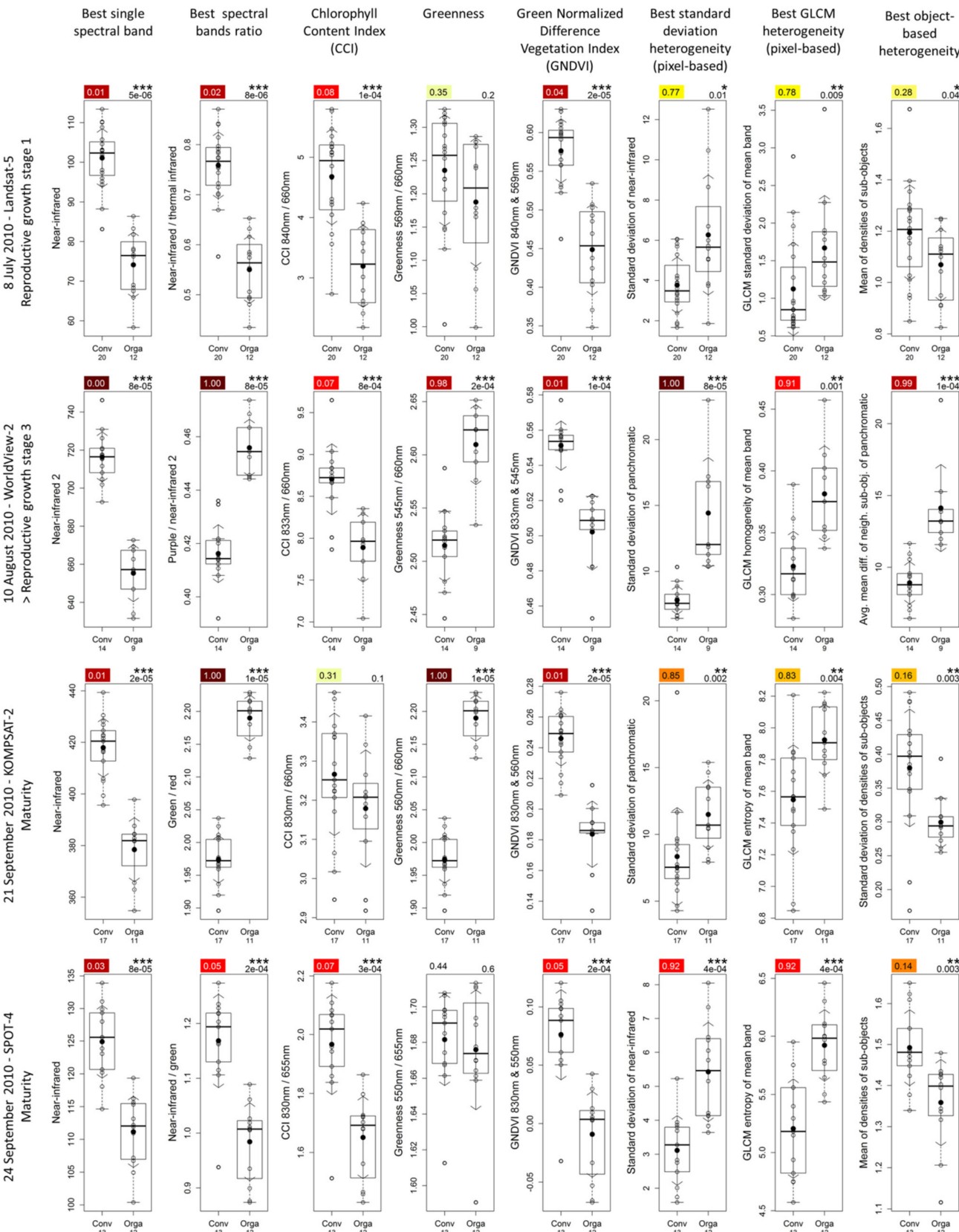

**Figure 9.** Selected indices computed from the four multispectral satellite images and providing the best discrimination between conventional and organic maize fields and related discrimination power expressed as the ROC-AUC (plot top-left value). Values of the spectral bands are Digital Numbers (DN). Detailed description of the plots available in Figure 4.

### 3.2.2. Chlorophyll Content Related Indices

Regarding the chlorophyll content related indices (Figures 8 and 9, columns 3-4-5), GNDVI and CCI indices presented always very significantly higher values for the conventional management mode and enabled a very high discrimination level in all situations, except for CCI on the KOMPSAT-2 image where the difference was not statistically significant. Close to complete separation was observed in most of the situations. The GNDVI always performed slightly better than the CCI and was very efficient for discrimination on the KOMPSAT-2 image whereas the CCI failed. These results are in line with the ones observed for the temporally corresponding in situ hyperspectral reflectance measurements (ASD 6th–9th August) and fully confirmed the study hypotheses that organic fields present lower chlorophyll content than conventional fields and that this can be observed from remote sensing.

The fact that the KOMPSAT-2 CCI was meaningfully less discriminant than the WorldView-2 (before) and SPOT-4 (three days later) can likely be partly explained by the difference in the number of fields involved in the analysis. The Greenness index was clearly less discriminant, presenting very high or perfect discrimination in two of the studied situations, and no statistically significant difference for the other two situations. This Greenness index behavior, when discriminant, with higher Greenness for organic fields, is also in line with the behavior observed for the temporally corresponding in situ hyperspectral reflectance measurements.

### 3.2.3. Spatial Heterogeneity Indices

All standard deviation (pixel-based) indices were higher for the organic fields (ROC-AUC > 0.5) (Figure 8). In particular, the most discriminant standard deviation index of each situation is from significantly to very significantly higher for organic fields and corresponds to moderate (Landsat-5), high (KOMPSAT-2), very high (SPOT-4), and complete (WorldView-2) discrimination levels (Figure 9, sixth column). This observation confirms the study hypotheses that organic fields present higher spatial heterogeneity compared with conventional fields and that it can be observed from remote sensing. The best standard deviation indices of each situation are either more or (near-) equally discriminant compared with best spatial heterogeneity of other types (GLCM and object-based). Depending on the situation, the standard deviation of other spectral bands may also provide high or very high discrimination levels between crop management modes (Figure 8).

The best GLCM spatial heterogeneity indices (pixel-based) showed a discrimination power very similar to one of best standard deviation indices (Figures 8 and 9, seventh column). However, we observed that the WorldView-2 image presented a lower spatial heterogeneity for organic fields corresponding to higher GLCM homogeneity index values, which is in opposition with the corresponding standard deviation index and temporal neighbors best GLCM indices that all showed a higher spatial heterogeneity for organic maize fields. Other GLCM indices for the WorldView-2 image (not presented) also presented a lower spatial heterogeneity for organic fields, which shows that this spatial heterogeneity inversion is not related to the GLCM homogeneity index itself.

The best object-based spatial heterogeneity indices showed high or very high discrimination levels for the three situations corresponding to images of higher spatial resolution, and a moderate discrimination power at the limit of the statistical significance for the lower spatial resolution Landsat-5 image (Figures 8 and 9, eighth column). Higher values of the index mean of densities of sub-objects, i.e., objects tending toward a square, corresponded to lower spatial heterogeneity (confer indices definition Table 2 page 12). We observed, on the KOMPSAT-2 image, that the standard deviation of the densities of sub-objects was higher for conventional fields, which may be related to the higher diversity of shapes of objects in conventional fields due to the higher visibility of tractor traces in these fields.

High or very high discrimination levels were achieved for all selected spatial heterogeneity indices corresponding to higher spatial resolution images (up to complete separation in one situation) acquired over a 1.5-month period, while the Landsat-5 image,

acquired earlier in the season, presented a generally lower, moderate discrimination power, which may be related to its relatively lower spatial resolution (30 m). However, the satellite image acquisition date and, consequently, the crop growth stage may be another factor strongly influencing the discrimination power of a given situation, along with the variable number of fields studied in each situation; therefore, no straight conclusion can be made from this last observation.

Finally, the vast majority of indices showed a higher spatial heterogeneity for the organic maize fields, except for the GLCM indices of the 10th of August and for the standard deviation of the densities of sub-objects index of the 21st of September. These last examples highlight the fact that different spatial heterogeneity indices express different types of spatial heterogeneity.

## 4. Discussion

### 4.1. Discussion on the In Situ Results

The in situ data analysis enabled us to show and quantify the biochemical and biophysical differences between organic and conventional maize. High and very high discrimination levels between management modes were achieved for many parameters, with up to perfect separation for the leaf hyperspectral reflectance (Figures 6 and 7).

The initial hypothesis that organic maize presents lower nitrogen and chlorophyll content than conventional maize was confirmed. Both maize leaf nitrogen content studied through laboratory analysis and maize leaf chlorophyll content studied in situ with a chlorophyll content meter showed very significantly lower values for organic maize. Although the CCI computed from the in situ hyperspectral reflectance showed no significant difference, the Greenness and GNDVI indices showed an excellent discrimination performance with nearly complete discrimination between management modes.

The initial hypothesis that organic maize presents lower field development (biomass and canopy cover) than conventional fields was confirmed. All parameters related to the field biomass and canopy cover, i.e., height, biomass weight, PAI, and FAPAR, showed very significantly lower values for organic fields and provided high to very high discrimination levels.

Regarding the initial hypothesis that organic maize presents higher spatial heterogeneity compared with conventional maize, while the maize height standard deviation by field showed significantly higher heterogeneity for organic maize, the standard deviation of the leaf chlorophyll content did not show any statistically significant difference. Whether the rather limited in situ sampling was sufficient to capture field spatial heterogeneity, which is typically expressed at the field scale, is questionable. For this reason, the spatial heterogeneity analysis based on in situ measurements does not enable us to conclude unequivocally in disfavor of the studied hypothesis.

The in situ maize leaf hyperspectral reflectance measurement enabled us to qualify and quantify the spectral differences between crop management modes, and, in particular, to identify the spectral wavelengths and their combinations enabling the best discrimination between crop management modes. The green and red-edge spectral regions were identified as the most discriminant. Regarding the index efficiency, while single spectral bands provided up to a very high discrimination level, the combination of two indices in the form of simple ratios was revealed, as expected, to be even better, equaling the most discriminant laboratory index, i.e., the leaf nitrogen content, and provided complete separation.

These conclusions are based on a spatially very limited in situ sampling corresponding to one single very short period (five days), and the discrimination power of the different indices analyzed may vary with the crop growth stages in a range that was not assessed.

Finally, we concluded that biochemical and biophysical differences between organic and conventional maize fields, induced by crop management modes, may exist at a certain time of the crop development, that these differences may be sufficiently pronounced to enable a complete discrimination between crop management modes, and that, given that the most discriminant indices found were from reflectance measurements/indices using

spectral bands typically available on common satellites, the later should also enable an efficient crop discrimination.

### 4.2. Discussion on the Satellite Results

The spectral range useful for crop management mode discrimination corresponds to the spectral range available on most common multispectral satellite sensors, such the green (alone or in CCI and GNDVI), red (mainly in CCI), red-edge, and NIR (alone or in CCI or GNDVI). Additional spectral ranges were revealed to also be discriminant depending on the situation; however, most of the time in combination with the aforementioned ones and without any real additional value.

The spatial resolution of the sensors impacted the different types of indices in different ways. Spectral indices do not require a specific spatial resolution to enable a very high discrimination. Regarding spatial heterogeneity indices, while various spatial resolutions (0.5–20 m) enabled high to full discrimination levels, the lowest spatial resolution image used (Landsat-5, 30 m) provided a significantly lower discrimination level. However, this last observation has to be balanced with the fact that the variation of both the number of fields and the maize growth stage studied on that lower spatial resolution image may also impact the observed discrimination level. The spatial resolution has to be adapted to the size of the fields analyzed to enable the computation of spectral indices from the pure field pixels and, if needed, to enable the extraction of a sufficient number of pixels through which field spatial heterogeneity can be expressed.

The spectral indices were revealed to be generally more discriminant than the spatial heterogeneity indices. Although spectral and spatial heterogeneity indices may be complementary as they express different field characteristics (for example crop vigor vs. crop spatial heterogeneity), it was clear that a single index type—for example, the NIR reflectance—may be sufficient to fully discriminate crop management modes. Similarly, a single image acquired at the right crop growth stage may be sufficient to achieve a full discrimination level. However, in situations other than those encountered in this study, discrimination might not be as easy and the different types of indices, as well as multiple dates of satellite image acquisitions, might be useful for efficient crop management mode discrimination.

Regarding the variation of the discrimination power of crop management modes with the studied crop growth stages, conclusions should be made with caution given that each of the studied situations differ in ways other than only the crop growth stage, i.e., the sensor spatial and spectral resolution and the number of fields studied. Even if certain spectral indices enabled very high discrimination throughout the different crop growth stages studied (NIR reflectance and GNDVI), other indices (CCI and especially the Greenness) were less stable and presented sensitivity to the maize growth stages.

This stable, very highly discriminant behavior observed for some indices over a long period of approximately 2.5 months is particularly interesting in the sense that, in the framework of an operational remote sensing supported organic crop certification process, it provides flexibility for the satellite image acquisitions and increases the probability to obtain usable cloud-free images during the period of interest. For spatial heterogeneity indices, no variation of the discrimination power could be attributed to the variation of crop growth stages specifically given the limitations mentioned previously.

Finally, we concluded that the initial study hypotheses that organic maize fields present lower chlorophyll content, lower crop biomass development, and higher spatial heterogeneity than conventional fields and that this can be observed from satellite imagery and used for crop management mode discrimination were fully confirmed for the first time. The results showed that both spectral and spatial heterogeneity indices derived from multispectral satellite imagery enabled very efficient to full discrimination between organic and conventional maize fields.

These results are consistent with those of Denis [14] showing that multispectral satellite sensors enabled full discrimination between organic or organic in conversion and

conventional wheat fields in a context similar to this study. The discrimination level achieved in this study was much higher than the one achieved by Denis and Tychon [15,16] for cotton fields in Burkina Faso, where both the important variation of intra and inter field spatial heterogeneity and the acquisition of a single satellite image very late in the crop cycle were suggested to be the main factors preventing a better discrimination level.

These results are very encouraging and suggest, for the first time, that satellite images could effectively support the organic maize certification process. However, given the limited representativeness of the dataset used in this study (approximately 30 maize fields located in a relatively small area) compared to the variety of situations in which this technique might potentially be applied, and in the view of developing an operational system, the next step is to test the robustness of the method at a larger scale in a more diverse context (larger dataset and another region and climate) and also for other crops. The method itself could also be enhanced with the use of multivariate-multitemporal discrimination techniques and with the use of more recent satellite sensors, such as Sentinel-1, Sentinel-2, and SkySat, and PlanetScope very high spatial resolution constellations.

## 5. Conclusions

This research demonstrated that highly significant biochemical and biophysical differences between a large number of organically and conventionally managed maize fields may exist at identified crop growth stages and that these differences may be sufficiently pronounced to enable a complete discrimination to be made between crop management modes using satellite images issued from common multispectral satellite sensors through the use of a range of spectral or spatial heterogeneity indices. These results are encouraging and suggest, for the first time, that satellite images could effectively support the organic maize certification process.

**Author Contributions:** Conceptualization, A.D., B.D., S.M., H.H., H.B., P.O. and B.T.; methodology, A.D.; software, A.D.; formal analysis, A.D.; investigation, A.D., B.D., S.M. and A.L.K.; writing— original draft, A.D.; writing—review and editing, A.D., B.D., S.M., H.H., H.B., P.O. and B.T.; supervision, H.H., H.B., P.O. and B.T.; project administration, H.H., H.B., P.O. and B.T.; funding acquisition, H.H., H.B., P.O. and B.T. All authors have read and agreed to the published version of the manuscript.

**Funding:** This research was funded by the European Space Agency (ESA/ESRIN) in 2010, as a User oriented project (12 months, 170,000 €, EOrganic project).

**Institutional Review Board Statement:** Not applicable.

**Informed Consent Statement:** Not applicable.

**Data Availability Statement:** The in situ datasets generated and/or analyzed during the current study and the satellite derived database are available from the corresponding author upon reasonable request. The geographical boundaries of the fields analyzed during the current study are not publicly available due to privacy protection. The satellite images analyzed during the current study are not publicly available due to restrictions from the commercial license, except for the Landsat-5 satellite image, which is freely available at https://earthexplorer.usgs.gov/.

**Acknowledgments:** For having supported this research, thanks to Yves CORNET, Antoine STEVENS, Bakary DJABY, Robert OGER, Jos VAN ORSHOVEN, Christian BARBIER, Christophe Mackels, Michel NOEL, Moussa EL JARROUDI, Catherine TIMMERMANS, the Centre de Recherche Public– Gabriel Lippmann (CRP-GL-Luxembourg). Field surveys benefited from full access to the fields and manpower and machinery support of the local farmer. Landsat-5 images courtesy of the U.S. Geological Survey.

**Conflicts of Interest:** The authors declare no conflict of interest. The funder had no role in the design of the study; in the collection, analyses, or interpretation of data; in the writing of the manuscript, or in the decision to publish the results.

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
