# Peer review of "Multispectral Remote Sensing as a Tool to Support Organic Crop Certification: Assessment of the Discrimination Level between Organic and Conventional Maize"

_remotesensing, doi:10.3390/rs13010117_

Round 1
Reviewer 1 Report
This paper suggest multispectral remote sensing images can be effectively used to discriminate organic and conventional maize. The paper contains results obtained by various spectral and non-spectral methods, which makes length of the research paper longer than it needs to be. Authors also needs proofreading and an extensive line-by-line editing for better readability. In general, the originality and experiment design is good, but there are issues that require authors' attention for improvement.
- Please check the title needs period at the end.
- In general, words should be used for numbers from zero through nine, and numerals should be used from 10 onwards. Use words for any number that is used to start a sentence, with the exception of years.
- There are very long noun clauses as subject or object, which is confusing to readers.
- Too long or choppy sentences reduces readability.
- There are paragraphs with only one sentence.
- Unnecessary quotation marks are found throughout the manuscript.
- Unnecessary use of square brackets are found throughout the manuscript.
- Line 92, e.g.:
- Line 100, please rephrase post-harvest-end-of-summer period.
- Please use superscript for -1 in ha-1
- Please consider using the word approximately instead of tilde.
- Please use multiplication sign instead of asterisk.
- Please use subscript for 2 in H2O.
- There is no information about the rightmost column in the Figure 3(f).
- Line 300, please briefly describe the necessity of calibration.
- Use subscript for i, j in Bi, Bj.
- In equation 4, does 931 nm mean reflectance at 931 nm?
- Line 310, acronym and longer name for CCI were presented earlier. Check for other words/phrases too.
- In general, it is common to use remote sensing data from a specific sensor/platform in multi-temporal analysis. What was the reason that you ended up using images from four different sensors/platforms?
- Use decimal point as decimal separator.
- Line 343, please rephrase satellite image spectral band.
- Line 368, an explanation is needed for parcels and sub-objects.
- Line 370, the meaning of the word "mean value of sub-objects" is not clear.
- Table 2, a brief explanation about superobject is required.
- Line 402, a brief explanation about statistic W is required.
- Figure 4, consider adding a, b, c, etc. as in Figure 6.
- Consider presenting only important results in Figure 4
- Section 3.1.2 can be merged to section 2.1.
- Figure 6, consider removing unnecessary results.
- Information missing for Figure 6(a-i).
- Line 447, rephrase maize 3-plants samples wet and dry weights. There are many long noun clauses like this.
- Figure 7(a), consider removing mean by field, or add range (or std) of mean by field spectral as shaded area.
- Figure 7(a), consider adding zoom-in plots in important wavelength regions you would like to highlight.
- Figure 7(b), consider adding Bi (nominator) and Bj (denominator) on the axes.
- Figure 7(b), please highlight important single band/combination of bands that has higher discrimination power.
- Line 524, Please rephrase following phrases: 1 situations, 1 field in 2 situations, 2 fields in 1 situation.
- Line 559, field survey was 6-10th from line 214.
- Line 569, ROC-AUC?
- Figure 9, NIR band shows best discrimination power in reproductive growth stage 3. What would be the reason the spectral response of two management modes do not show difference in 7(a)?
- Figure 9, the discrimination power when using single band was significantly higher compared to when using spectral/spatial heterogeneity were used. What would be advantages of using spectral/spatial indices? Please consider this is very important question that can justify the need of spectral/spatial metrics to answer your research question.
- Figure 7 and 9, discrimination power of CCI in reproductive growth stage 3 seems quite different. What could be the reason?
- Please find better word/phrase for situation.
- Line 631, what could be the reason that CCI computed from hyperspectral data does not have high discrimination power, whereas the result from chlorophyll meter showed significantly lower value in organic maize.
- Line 663, 4.2.
- Is there a possibility that a conventional plot with a disease or poorly-managed plot misclassified to organic field?
- Line 275, use multiplication sign instead of alphabet x.
Reviewer 2 Report
The authors of this article performed an interesting work in the elaboration of a methodology that allowed the differentiation between conventional and crop fields declared as organic using spatial remote sensing data with the support of in situ measurements (leaf hyperspectral reflectance, chlorophyll and nitrogen content and dry matter percentage, crop canopy cover, height, wet biomass and dry matter percentage, soil chemical composition) which were in agreement the spatial remote sensing data.
Some comments that may improve the article:
- Line 781: please correct the name of the congress
- In Figure 1: a presentation of the global applied methodology was presented but “the discriminant power matrix” was not described in the methodology part.
- In Figure 2: I think it will be better if the map of the whole country will be included in small box in the bottom right or left corner to give more visibility to the study area, and the plots.
